# Preparation of Chiral Enantioenriched Densely Substituted Cyclopropyl Azoles, Amines, and Ethers via Formal S*_N_*2′ Substitution of Bromocylopropanes

**DOI:** 10.3390/molecules27207069

**Published:** 2022-10-20

**Authors:** Hillary Straub, Pavel Ryabchuk, Marina Rubina, Michael Rubin

**Affiliations:** 1Department of Chemistry, University of Kansas, Lawrence, KS 66045, USA; 2Department of Chemistry, North Caucasus Federal University, 355009 Stavropol, Russia

**Keywords:** cyclopropenes, cyclopropanes, nucleophilic addition, metal-templated reactions

## Abstract

Enantiomerically enriched cyclopropyl ethers, amines, and cyclopropylazole derivatives possessing three stereogenic carbon atoms in a small cycle are obtained via the diastereoselective, formal nucleophilic substitution of chiral, non-racemic bromocyclopropanes. The key feature of this methodology is the utilization of the chiral center of the cyclopropene intermediate, which governs the configuration of the two adjacent stereocenters that are successively installed via 1,4-addition/epimerization sequence.

## 1. Introduction

Enantiomerically pure cyclopropane derivatives are ubiquitous, nature inspired [1,2,3,4,5,6,7,8] building blocks abundantly employed in organic synthesis [9,10,11,12,13], asymmetric catalysis [14,15,16,17], and medicinal chemistry [18,19,20,21,22,23,24,25]. These advanced synthons are typically accessed via diastereoselective 1,3-ring closure reactions [26,27,28,29,30] or asymmetric cyclopropanation [31,32,33,34,35,36,37,38,39,40,41]. A less established, complementary approach relies on chemo- and diastereoselective installation of additional substituents into pre-formed chiral or prochiral cyclopropanes [42,43,44,45]. Strain-release-driven additions of different entities to cyclopropenes proved useful for the assembly of enantiomerically enriched cyclopropane derivatives that are not easily accessible via other methods [46,47,48,49,50,51,52,53]. Synthetic methodologies exploiting stereoselective ring-retentive, metal-catalyzed [13,54,55], and organocatalytic [56,57] additions to cyclopropenes were developed by several research groups and have eventually evolved into a rapidly growing area. Our group recently disclosed an efficient diastereoselective route to cyclopropanes **3** via a formal substitution of bromocyclopropanes **1** with oxygen, nitrogen, or sulfur-based nucleophiles (Figure 1) [51,58]. The reaction proceeds via a base-assisted dehydrohalogenation, affording a highly reactive cyclopropene intermediate **2** and the subsequent nucleophilic addition across the double bond of cyclopropene. Herein, we report our progress on extending this methodology for the preparation of enantiomerically enriched cyclopropanes. 

## 2. Results and Discussion

This section may be divided by subheadings. It should provide a concise and precise description of the experimental results, their interpretation, and the experimental conclusions that can be drawn. In our earlier studies of the formal nucleophilic substitution of bromocyclopropanes, we have demonstrated several reaction modes that allow for efficient control of the diastereoselectivity of this transformation (Figure 1). Thus, it was shown that the derivatives of 2-bromocyclopropylcarboxylic acid **4** produced achiral cyclopropene **5** upon treatment with base. The latter underwent in situ addition of nucleophiles to afford *trans*-cyclopropane **6**. The high diastereoselectivity of the addition was attributed to a base-assisted, thermodynamically driven epimerization of the tertiary carbon atom (C-1, mode **A**, Figure 1) [59,60,61,62,63]. Alternative approaches to control the diastereoselectivity of the intermolecular nucleophilic substitution were also developed by utilizing 1,2,2-trisubstituted cyclopropanes as the starting materials. The first approach employs substrates bearing two substituents with significantly different steric demands (**7**, small R_S_, and large R_L_). The in situ generated achiral cyclopropene **8** undergoes nucleophilic attack at the least hindered face, resulting in selective formation of product **9** (Figure 1, mode **B**) [62,63]. The second approach takes advantage of bromocyclopropane **10**, bearing a directing functionality (DG, typically carboxamide or carboxylic acid group) capable of efficient coordination to the potassium cation, which serves as a delivery vehicle for the nucleophilic counter-anion. Overall, the addition to the double bond of cyclopropene **11** proceeds in cis fashion, with respect to the directing functional group furnishing **12** with high diastereoselectivity (Figure 1, mode **C**) [62,63]. The addition of a tethered alkoxide entity was also investigated; both *exo-trig* (Figure 1, mode **D**) [64,65] and *endo*-*trig* (mode **E**) [66] modes efficiently provided the corresponding medium-size heterocycles **15** and **18**.

Attempts to extend this methodology beyond the trisubstituted cyclopropane substrates greatly amplify the challenge of controlling the stereoselectivity of the addition. Indeed, all the modes discussed above require the control of a single center only, since the two forming chiral centers are linked to each other. In 2013, we communicated on the realization of a more advanced strategy, involving two modes of diastereoselctivity control providing tetrasubstituted cyclopropyl ethers **23** (mode **F**, Figure 2) [67]. The proof of concept of such a strategy was showcased on racemic bromocyclorporpanes **20**. We also demonstrated employing racemic substrates, i.e., that the relative configuration of the center at C-3 can be efficiently controlled by steric environment employing appropriate substituents R_S_, R_L_; thus, control of this step is related to mode **B**. Finally, relative configuration at C-1 was installed via the base-assisted epimerization of this center, in a process identical to the one, previously used in mode **A** (Figure 2) [66,67]. We reasoned that the absolute configuration of the quaternary stereogenic center at C-2 in chiral non-racemic amide **20** would be preserved during the dehydrohalogenation/nucleophilic addition sequence, which can be used to access to enantiomerically enriched compounds **23**. 

In order to access the densely substituted enantiopure cyclopropanes, we have developed a very facile protocol for the chiral resolution of carboxylic acids **19**, utilizing the re-crystallization of racemic acids with cinchona alkaloids [68]. It was shown that a variety of enantiomerically enriched acids **19** with ee > 95% were available in multi-gram scale, in both enantiomeric forms after single crystallization of either cinchonine or cinchonidine salts. Enantiopure acids can easily be converted into amides **20**, as a precursor for enantioenriched cyclopropenes (Figure 3). 

### 2.1. Alcohol Nucleophiles

With enantiomerically pure amides in hand, we have utilized the “dual control” mode of the formal nucleophilic substitution of bromide with various alkoxides (Figure 4). At 40 °C in DMSO, bromocyclopropanes **20** were converted to the corresponding cyclopropanes **23**. Primary alcohols served as excellent nucleophiles for the title reaction, with diastereoselectivity greater than 25:1 in all cases. We demonstrated that this methodology is complementary to our previous report, providing an easy access to the enantiopure cyclopropyl ethers.

### 2.2. Azole Nucleophiles

The nitrogen-based nucleophiles in our original report have been explored to a lesser extent. Therefore, we became interested in utilizing homochiral amides **20** for the generation of the corresponding cyclopropyl amines. We tested a series of different amines as N-pronucleophiles; however, our initial attempts to induce the addition primary and secondary alkyl amines, as well as carboxamides and sulfonamides, were unsuccessful. We were pleased to find that the azoles underwent a facile addition to cyclopropenes to provide substituted hetarylcyclopropanes in an optically pure form (Figure 5). The reaction in the presence of pyrrole afforded the corresponding tetrasubstituted cyclopropanes (+)-**23cag** and (+)-**23dag** in high yields and with excellent diastereoselectivities. We were glad to find that such problematic nucleophiles, such as indoles, known for their susceptibility to Friedel–Crafts alkylation, dimerization, and polymerization, afforded good, isolated yields of the corresponding adducts. Substituted indoles and 7-azaindole proceeded cleanly to afford the corresponding cyclopropanes. Similarly, pyrazole was engaged in a very efficient transformation with enantiomerically pure cyclopropyl bromide, providing (+)-**23cah** in good, isolated yield, although longer reaction times were reacquired, and the diastereoselectivity was slightly lower. More acidic azoles, including imidazoles, benzimidazoles, and triazoles, did not participate in the title reaction, due to deactivation of the base in the reaction media, thus preventing the generation of the cyclopropane intermediate. The sensitivity of the reaction to sterics can be seen by comparing the reactivity of bromocyclopropanes possessing a methyl and an ethyl group, respectively, at the β-quaternary center. Compared to methyl-tolyl cyclopropane (+)-**23cag**, its ethyl/phenyl isomer reacted very sluggishly at 40 °C and required higher temperature to achieve full conversion, which led to a lower, although still respectable, diastereoselectivity of 15:1 for **23aag**. To our delight, the carboxamide, possessing a larger naphthyl substituent, also participated in the substitution reaction with pyrrole, giving (+)-**23dag** as a single enantiomer. 

### 2.3. Aniline Nucleophiles

Anilines were also tested in this reaction, and, to our delight, *N*-methylaniline gave a cyclopropyl amine (+)-**23ack** in 55% and dr 3:1. *p*-Flouro-N-methylaniline can be utilized in the described reaction, providing a tetrasubstituted cyclopropane (+)-**23acm** with similar diastereoselectivity and yield. It was found that increased steric hindrance at the N-termini of the pronucleophile had a significant effect on the reaction course. Thus, aniline bearing an ethyl substituent greatly increased the reaction’s efficacy; the diastereoselectivity increased to 13:1 for (+)-**23acl**. The utilization of naphtyl-substituted bromocyclopropane precursor gave exclusive formation of (+)-**23dal** (Figure 5). Unfortunately, anilines with the secondary alkyl group at the nitrogen atom did not participate in this reaction, most likely due to the excessive steric demands (compound **23ddl** in Figure 5).

## 3. Materials and Methods

### 3.1. General 

NMR spectra (See Appendix A) were recorded on a Bruker Avance DRX-500 (500 MHz) with a dual carbon/proton cryoprobe (CPDUL). ^13^C NMR spectra were registered with broadband decoupling. The (+) and (−) designations represent positive and negative intensities of signals in ^13^C DEPT-135 experiments. Numbers of magnetically equivalent carbons for each signal in ^13^C NMR spectra (unless it is one) are also reported. IR spectra were recorded on a ThermoFisher Nicolet iS 5 FT-IR Spectrometer. HRMS was carried out on LCT Premier (Micromass Technologies) instrument, employing ESI TOF detection techniques. Glassware used in moisture-free syntheses was flame-dried in vacuum prior to use. Column chromatography was carried out on silica gel (Sorbent Technologies, 40–63 mm). Precoated silica gel plates (Sorbent Technologies Silica XG 200 mm) were used for TLC analyses. Anhydrous dichloromethane was obtained by passing degassed commercially available HPLC-grade inhibitor-free solvent consecutively through two columns filled with activated alumina and stored over molecular sieves under nitrogen. Water was purified by dual stage deionization, followed by dual stage reverse osmosis. Anhydrous THF was obtained by refluxing commercially available solvent over calcium hydride, followed by distillation in a stream of dry nitrogen. All other reagents and solvents were purchased from commercial vendors and used as received. Diastereomeric ratios of products were measured by GC and NMR analyses of crude reaction mixtures. In the event where minor diastereomer was not detected by either of these methods, ratio >100:1 was reported. 

### 3.2. Preparation of Starting Materials

***(+)-(1S,2R)-1-Bromo-N-(tert-butyl)-2-methyl-2-phenylcyclopropane-1-carboxamide*** (**20aa**). Typical procedure A. A flame dried 100 mL round-bottom flack, equipped with drying tube and magnetic stir bar, was charged with (1*S*,2*R*)-1-bromo-2-methyl-2-phenylcyclopropane carboxylic acid (**19a**) (1.10 g, 4.33 mmol, 1.00 equiv.), DMF (10 mL), and anhydrous dichloromethane (40 mL). The mixture was treated with oxalyl chloride (563 μL, 6.50 mmol, 1.50 equiv.) at 0 °C, stirred for 15 min, warmed to room temperature, and additionally stirred for 2 h. The solvent was removed in vacuum, and the crude acyl chloride was dissolved in dry THF (20 mL), followed by addition of a solution of *tert*-butyl amine (**24a**) (1.35 mL, 12.8 mmol, 2.97 equiv.) in THF (20 mL). The reaction mixture was stirred overnight. After the reaction was complete, the solvent was removed in vacuum, and the residue was partitioned between EtOAc (25 mL) and water (25 mL). The organic phase was separated, and the aqueous layer was extracted with EtOAc (2 × 25 mL). The combined organic phases were dried (MgSO_4_), filtered, and concentrated. The residual crude oil was purified by column chromatography on silica gel. The titled compound obtained a colorless solid, mp: 83.2–86.0 °C, R_ƒ_ 0.55 (hexanes/EtOAc 6:1), [α]_D_ = +14.0° (c 0.172, CH_2_Cl_2_). Yield 1.03 g (3.32 mmol, 77%). Spectral properties of this material were identical to those reported for the racemic amide [67]. 

***(+)-(1R,2S)-1-Bromo-N,N,2-trimethyl-2-phenylcyclopropane-1-carboxamide*** (**20ab**). Compound was obtained via typical procedure A, employing (1*R*,2*S*)-1-bromo-2-methyl-2-phenylcyclopropane-1-carboxylic acid (**19a**) (510 mg, 2.01 mmol, 1.00 equiv.), oxalyl chloride (260 μL, 3.03 mmol, 1.51 equiv), and 40 wt.% aq solution of dimethyl amine (**24b**) (753 μL, 8.91 mmol, 4.43 equiv.). Chromatographic purification afforded title compound as a colorless solid, mp: 81.7–83.3 °C, R_ƒ_ 0.34 (hexanes/EtOAc 10:1), [α]_D_ = +13.3° (c 0.098, CH_2_Cl_2_). Yield 466 mg (1.66 mmol, 83%). ^1^H NMR (500 MHz, CDCl_3_) *δ*_H_ 7.43–7.10 (m, 5H), 2.65 (s, 3H), 2.57 (d, ^2^*J*_H,H_ = 7.4 Hz, 1H), 2.56 (s, 3H) 1.85 (s, 3H), 1.37 (d, ^2^*J*_H,H_ = 7.4 Hz, 1H); ^13^C NMR (126 MHz, CDCl_3_) *δ*_C_ 166.5, 138.2, 128.1 (+, 2C), 127.1 (+, 2C), 126.3, 42.6, 38.5 (+), 30.8, 27.0 (-), 24.1 (+); FT IR (KBr, cm^−1^): 2927, 1647, 1558, 1496, 1396, 1272, 1176, 1082, 1058, 1029, 954, 763, 696, 680, 669, 650; HRMS (TOF ES): found 281.0415, calculated for C_13_H_16_BrNO (M^+^) 281.0415 (0.0 ppm). 

***(+)-(1S,2R)-1-Bromo-N,N-diethyl-2-methyl-2-phenylcyclopropane-1-carboxamide*** (**20ac**). Compound was obtained via typical procedure A, employing (1*S*,2*R*)-1-bromo-2-methyl-2-phenylcyclopropane-1-carboxylic acid (**19a**) (515 mg, 2.02 mmol, 1.00 equiv.), oxalyl chloride (260 μL, 3.03 mmol, 1.50 equiv.), and diethyl amine (**24c**) (823 μL, 7.96 mmol, 3.96 equiv.). Chromatographic purification afforded title compound as a light yellow solid, mp: 74.3–76.2 °C, R_ƒ_ 0.34 (hexanes/EtOAc 10:1), [α]^25^_D_ = +17.0° (c 0.194, CH_2_Cl_2_). Yield 537 mg (1.74 mmol, 86%). ^1^H NMR (500 MHz, CDCl_3_) *δ*_H_ 7.43–7.04 (m, 5H), 3.48 (dq, ^2^*J*_H,H_ = 14.2, ^3^*J*_H,H_ = 7.1 Hz, 1H), 3.37 (dq, ^2^*J*_H,H_ = 14.2, ^3^*J*_H,H_ = 7.1 Hz, 1H), 2.71 (dq, ^2^*J*_H,H_ = 14.0, ^3^*J*_H,H_ = 7.1 Hz, 1H) 2.65 (d, ^2^*J*_H,H_ = 7.3 Hz, 1H), 2.58 (dq, ^2^*J*_H,H_ = 14.0, ^3^*J*_H,H_ = 7.0 Hz, 1H), 1.85 (s, 3H), 1.33 (d, ^2^*J*_H,H_ = 7.3 Hz, 1H), 1.0 (t, ^3^*J*_H,H_ = 7.1 Hz, 3H) 0.47 (t, ^3^*J*_H,H_ = 7.1 Hz, 3H); ^13^C NMR (126 MHz, CDCl_3_) *δ*_C_ 165.8, 138.1, 128.1 (+), 127.0 (+), 126.7 (+), 42.4, 42.0 (-), 38.3 (-), 31.0, 26.9 (-), 24.3 (+), 12.6 (+), 11.1 (+); FT IR (KBr, cm^−1^): 2977, 2933, 1643, 1639, 1498, 1456, 1433, 1380, 1282, 1219, 1064, 719, 696, 582; HRMS (TOF ES): found 309.0724, calculated for C_15_H_20_BrNO (M^+^) 309.0728 (1.3 ppm). 

***(-)-(1S,2R)-1-Bromo-N-benzyl-1-bromo-2-methyl-2-phenylcyclopropane-1-carboxamide*** (**20ad**). Compound was obtained via typical procedure A, employing (1*S*,2*R*)-1-bromo-2-methyl-2-phenylcyclopropane carboxylic acid (**19a**) (255 mg, 1.00 mmol, 1.00 equiv.), oxalyl chloride (130 μL, 1.50 mmol, 1.50 equiv.), and benzyl amine (**24d**) (327 μL, 3.00 mmol, 3.0 equiv). Chromatographic purification afforded a colorless solid, mp: 88.2-91.3 °C, R_ƒ_ 0.36 (hexanes/EtOAc 6:1), [α]^25^_D_ = −115.2° (c 0.046, CH_2_Cl_2_). Yield 240 mg (0.700 mmol, 70%). Spectral properties of this material were identical to those reported for the racemic amide [67]. 

***(+)-(1S,2R)-1-Bromo-2-methyl-2-phenylcyclopropyl)(pyrrolidin-1-yl)methanone*** (**20ae**). Compound was obtained via typical procedure A, employing (1*S*,2*R*)-1-bromo-2-methyl-2-phenylcyclopropane carboxylic acid (**19a**) (255 mg, 1.00 mmol, 1.00 equiv.), oxalyl chloride (130 μL, 1.50 mmol, 1.50 equiv.) and pyrrolidine (**24e**) (246 μL, 3.00 mmol, 3.00 equiv). Chromatographic purification afforded a colorless oil, R_ƒ_ 0.39 (hexanes/EtOAc, 3:1), [α]^25^_D_ = +12.5° (c 0.172, CH_2_Cl_2_). Yield 289 mg (0.941 mmol, 94%). Spectral properties of this material were identical to those reported for the racemic amide [67]. 

***(+)-(1R,2S)-1-Bromo-N-(tert-butyl)-2-ethyl-2-phenylcyclopropane-1-carboxamide*** (**20ba**). Compound was obtained via typical procedure A, employing (1*R*,2*S*)-1-bromo-2-ethyl-2-phenylcyclopropane carboxylic acid (**19b**) (1.00 g, 3.73 mmol, 1.00 equiv.), oxalyl chloride (711 μL, 5.60 mmol, 1.50 equiv.) and *tert*-butyl amine (**24a**) (1.18 mL, 11.2 mmol, 3.00 equiv.). Chromatographic purification afforded a colorless solid, mp: 63.8–65.7 °C, R_ƒ_ 0.52 (hexanes/EtOAc 9:1), [α]^25^_D_ = +5.8° (c 0.052, CH_2_Cl_2_). Yield 1.06 g (3.27 mmol, 88%). Spectral properties of this material were identical to those reported for the racemic amide [67]. 

***(+)-(1S,2R)-1-Bromo-N-(tert-butyl)-2-methyl-2-(p-tolyl)cyclopropane-1-carboxamide*** (**20ca**). Compound was obtained via typical procedure A (1*S*,2*R*)-1-bromo-2-methyl-2-(*p*-tolyl)cyclopropane-1-carboxylic acid (**19c**) (240 mg, 0.89 mmol, 1.00 equiv.), oxalyl chloride (116 μL, 1.35 mmol, 1.52 equiv.) and *tert*-butyl amine (**24a**) (280 μL, 2.67 mmol, 3.00 equiv.). Chromatographic purification afforded a colorless solid, mp: 78.4–81.2 °C, R_ƒ_ 0.35 (hexanes/EtOAc 20:1), [α]^25^_D_ = +14.3° (c 0.071, CH_2_Cl_2_). Yield 225 mg (0.694 mmol, 78%). Spectral properties of this material were identical to those reported previously [67]. 

***(-)-(1R,2S)-1-Bromo-N-(tert-butyl)-2-methyl-2-(naphthalen-2-yl)cyclopropane-1-carboxamide*** (**20da**). Compound was obtained via typical procedure A, employing (1*R*,2*S*)-1-bromo-2-methyl-2-naphthalen-2-yl)cyclopropane-1-carboxylic acid (**19d**) (608 mg, 1.99 mmol, 1.00 equiv.), oxalyl chloride (260 μL, 3.00 mmol, 1.51 equiv.), and *tert*-butyl amine (**24a**) (630 μL, 6.00 mmol, 3.02 equiv). Chromatographic purification afforded title compound as a colorless solid, mp: 88.1–89.6 °C, R_ƒ_ 0.32 (hexanes/EtOAc 20:1), [α]^25^_D_ = −41.9° (c 0.418, CH_2_Cl_2_). Yield 427 mg (1.19 mmol, 60%). ^1^H NMR (400 MHz, CDCl_3_) *δ*_H_ 7.82–7.68 (m, 3H), 7.65 (d, ^3^*J*_H,H_ = 1.2 Hz, 1H), 7.46–7.39 (m, 2H) 7.35 (dd, ^3^*J*_H,H_ = 8.5, ^4^*J*_H,H_ = 1.8 Hz, 1H), 6.32 (br. s, 1H), 2.69 (d, ^2^*J*_H,H_ = 6.3 Hz, 1H), 1.78 (s, 3H), 1.34 (d, ^2^*J*_H,H_ = 6.3 Hz, 1H), 1.03 (s, 9H); ^13^C NMR (126 MHz, CDCl_3_) *δ*_C_ 165.3, 138.2, 133.3, 132,5, 128.0 (+), 127.8 (+), 127.7 (+), 126.8 (+) 126.3 (+), 126.1 (+), 125.7 (+), 51.6, 45.2, 35.1, 28.4 (+, 3C), 28.1 (+), 26.6(-); FT IR (KBr, cm^−1^): 3421, 3053, 2964, 2925, 1678,1599, 1512, 1454, 1392, 1363, 1290, 1221, 1134, 1063, 958, 893, 856, 815, 750; HRMS (TOF ES): found 359.0883, calculated for C_19_H_22_BrNO (M^+^) 359.0885 (0.6 ppm). 

### 3.3. Nucleophilic Addition Reactions

**(+)-(1*R*,2*R*,3*S*)-3-(Benzyloxy)-*N*-(*tert*-butyl)-2-methyl-2-phenylcyclopropane-1-carboxamide** (**23aaf**). Typical procedure B. An oven-dried 10 mL Weaton vial was charged with 18-crown-6 ether (5.3 mg, 20 μmol, 10 mol%), t-BuOK (134 mg, 1.20 mmol, 6.00 equiv.), benzyl alcohol (**25f**) (62.2 μL, 0.598 mmol, 2.92 equiv.), and anhydrous DMSO (10.0 mL). The mixture was stirred at room temperature for 1 min, and (1*S*,2*R*)-1-bromo-*N*-(*tert*-butyl)-2-methyl-2-phenylcyclopropane-1-carboxamide (**20aa**) 63.5 mg (0.205 mmol, 1.00 equiv.) was added in single portion. The reaction mixture was stirred overnight at 80 °C, then solvent was removed in vacuum, and the residue was partitioned between water (15 mL) and EtOAc (15 mL). The organic layer was separated, and the aqueous phase was extracted with EtOAc (3 × 15 mL). Combined organic extracts were washed with brine, dried over MgSO_4_, filtered, and evaporated. Flash column chromatography on silica gel afforded the titled compound as a colorless solid, mp: 139.9–142.3 °C; R_ƒ_ 0.32 (hexanes/EtOAc 4:1), [α]^25^_D_ = +14.5° (c 0.076, CH_2_Cl_2_). dr 60:1. Yield 52.0 mg (0.154 mmol, 76%). Spectral properties of this material were identical to those reported earlier for the racemic compound [67]. 

***(+)-(1R,2R,3S)-N-(tert-Butyl)-3-methoxy-2-methyl-2-phenylcyclopropane-1-carboxamide*** (**23aaa**). Compound was obtained according to typical procedure B from 65.6 mg (0.211 mmol, 1.00 equiv.) of (1*S*,2*R*)-1-bromo-*N*-(*tert*-butyl)-2-methyl-2-phenylcyclopropane-1-carboxamide (**20aa**) employing methanol (**25a**) (24.2 μL, 0.598 mmol, 2.84 equiv.) as pronucleophile. Chromatographic purification afforded 53.1 mg (0.197 mmol, 93%) of the title compound as a colorless solid, mp: 120.6-122.9 °C; R_ƒ_ 0.26 (hexanes/EtOAc 3:1), [α]^25^_D_ = +17.6° (c 0.068, CH_2_Cl_2_). dr 42:1. Spectral properties of this material were identical to those reported for the racemic compound [67]. 

***(+)-(1R,2R,3S)-N-(tert-Butyl)-3-ethoxy-2-methyl-2-phenylcyclopropane-1-carboxamide*** (**23aab**). Compound was obtained according to typical procedure B from 63.2 mg (0.204 mmol, 1.00 equiv.) of (1*S*,2*R*)-1-bromo-*N*-(*tert*-butyl)-2-methyl-2-phenylcyclopropane-1-carboxamide (**20aa**), employing ethanol (**25b**) (35.0 μL, 0.600 mmol, 2.95 equiv.) as pronucleophile. Chromatographic purification afforded 42.3 mg (0.155 mmol, 78%) of the title compound as a white solid, mp: 130.4–131.6 °C; R_ƒ_ 0.33 (hexanes/EtOAc 3:1), [α]^25^_D_ = +10.5° (c 0.048, CH_2_Cl_2_). dr 30:1. Spectral properties of this material were identical to those reported for the racemic compound [67].

***(-)-(1R,2R,3S)-N-(tert-Butyl)-2-methyl-2-phenyl-3-propoxycyclopropane-1-carboxamide*** (**23aac**). Compound was obtained according to typical procedure B from 62.0 mg (0.200 mmol, 1.00 equiv.) of (1*S*,2*R*)-1-bromo-*N*-(*tert*-butyl)-2-methyl-2-phenylcyclopropane-1-carboxamide (**20aa**), employing *n*-propanol (**25c**) (44.9 μL, 0.600 mmol, 3.00 equiv.) as pronucleophile. Chromatographic purification afforded 52.9 mg (0.182 mmol, 91%) of the title compound as a colorless solid, mp: 122.1–124.9 °C; R_ƒ_ 0.37 (hexanes/EtOAc 3:1), [α]^25^_D_ = −25.0° (c 0.044, CH_2_Cl_2_). dr 44:1. Spectral properties of this material were identical to those reported for the racemic compound [67]. 

***(+)-(1R,2R,3S)-N-(tert-Butyl)-3-(2-methoxyethoxy)-2-methyl-2-phenylcyclopropane-1-carboxamide*** (**23aae**). Compound was obtained according to typical procedure B from 63.5 mg (0.205 mmol, 1.00 equi v.) of (1*S*,2*R*)-1-bromo-*N*-(*tert*-butyl)-2-methyl-2-phenylcyclopropane-1-carboxamide (**20aa**), employing 2-methoxyethanol (**25e**) (47.3 μL, 0.601 mmol, 2.93 equiv.) as pronucleophile. Chromatographic purification afforded 46.1 mg (0.152 mmol, 76%) of the title compound as a colorless solid, mp: 101.9–104.2 °C; R_ƒ_ 0.34 (hexanes/EtOAc 1:1), [α]^25^_D_ = +70.0° (c 0.056, CH_2_Cl_2_). dr 58:1. Spectral properties of this material were identical to those reported for the racemic compound [67]. 

**(-)-(1*R*,2*R*,3*S*)-3-Methoxy-*N*,*N*,2-trimethyl-2-phenylcyclopropane-1-carboxamide** (**23aba**). Compound was obtained according to typical procedure B from 56.4 mg (0.201 mmol, 1.00 equiv.) of (1*S*,2*R*)-1-bromo-*N*,*N*,2-trimethyl-2-phenylcyclopropane-1-carboxamide (**20ab**), employing methanol (25.8 μL, 0.633 mmol, 3.15 equiv.) as pronucleophile. Chromatographic purification afforded 31.2 mg (0.134 mmol, 67%) as a colorless solid, mp: 108.9–111.3 °C, R_ƒ_ 0.28 (hexanes/EtOAc 1:1), [α]^25^_D_ = −96.6° (c 0.089, CH_2_Cl_2_). dr >100:1. ^1^H NMR (500 MHz, CDCl_3_) *δ*_H_ 7.34–7.12 (m, 5H), 4.25 (d, ^3^*J*_H,H_ = 5.8 Hz, 1H), 4.07 (s, 3H), 3.55 (s, 3H), 2.83 (s, 3H), 2.00 (d, ^3^*J*_H,H_ = 5.8 Hz, 1H), 1.65 (s, 3H); ^13^C NMR (126 MHz, CDCl_3_) *δ*_C_ 167.5, 140.2, 127,4 (+, 2C), 126,9 (+, 2C), 125.7 (+), 66.8 (+), 57.3 (+), 36.3 (+), 36.0, 34.2 (+), 33.3 (+), 19.6 (+); FT IR (KBr, cm^−1^): 2972, 2929, 1734, 1637, 1479, 1460, 1430, 1377, 1265, 1230, 1143, 1070, 952, 847, 796, 763, 759, 740, 698, 624; HRMS (TOF ES): found 234.1498, calculated for C_14_H_20_NO_2_ (M+H)^+^ 234.1494 (1.7 ppm). 

***(-)-(1S,2S,3R)-N,N-Diethyl-2-methyl-3-(3-methyl-1H-indol-1-yl)-2-phenylcyclopropane-1-carboxamide*** (**23aci**). Compound was obtained according to typical procedure B from 62.0 mg (0.201 mmol, 1.00 equiv.) of (1*R*,2*S*)-1-bromo-*N*,*N*-diethyl-2-methyl-2-phenylcyclopropane-1-carboxamide (**20ac**), employing skatole (**25i**) (79.0 mg, 0.602 mmol, 3.00 equiv.) as pronucleophile. Chromatographic purification afforded 49.6 mg (0.138 mmol, 69%) as a colorless solid, mp: 116.2–117.5 °C, R_ƒ_ 0.21 (hexanes/EtOAc 5:1), [α]^25^_D_ = −64.0° (c 0.050, CH_2_Cl_2_). dr 32:1. ^1^H NMR (500 MHz, CDCl_3_) *δ*_H_ 7.58 (d, ^3^*J*_H,H_ = 7.8 Hz, 1H), 7.48 (d, ^3^*J*_H,H_ = 8.1 Hz, 1H), 7.41 (d, ^3^*J*_H,H_ = 7.8 Hz, 2H), 7.27 (d, ^3^*J*_H,H_ = 6.0 Hz, 1H), 7.24–7.20 (m, 1H), 7.14 (dd, ^3^*J*_H,H_ = 11.0, 3.9 Hz, 1H), 6.92 (s, 1H), 4.63 (d, ^3^*J*_H,H_ = 4.2 Hz, 1H), 3.76 (td, ^2^*J*_H,H_ = 14.3, ^3^*J*_H,H_ = 7.0 Hz, 1H), 3.66 (td, ^2^*J*_H,H_ = 13.9, ^3^*J*_H,H_ = 7.0 Hz, 1H), 3.34 (dq, ^2^*J*_H,H_ = 14.3, ^3^*J*_H,H_ = 7.0 Hz, 1H), 2.93 (dq, ^2^*J*_H,H_ = 13.9, ^3^*J*_H,H_ = 7.0 Hz, 1H), 2.55 (d, ^3^*J*_H,H_ = 4.2 Hz, 1H), 2.34 (s, 3H), 1.42 (s, 3H), 1.32 (t, ^3^*J*_H,H_ = 7.1 Hz, 3H), 0.86 (t, ^3^*J*_H,H_ = 7.1 Hz, 3H); ^13^C NMR (126 MHz, CDCl_3_) *δ*_C_ 166.5, 140.3, 137.9, 129.4, 128.7, (+, 2C), 128.1 (+, 2C), 127.2 (+), 125.7 (+), 122.0 (+), 119.3 (+), 119.2 (+), 111.1, 110.5 (+), 42.3 (+), 41.9 (-), 40.2 (-), 37.0, 35.9 (+), 22.1 (+), 15.9 (+), 12.7 (+), 9.8 (+); FT IR (KBr, cm^−1^): 2972, 2927, 1639, 1465, 1379, 1309, 1263, 1230, 1143, 759, 740, 698; HRMS (TOF ES): found 360.2200, calculated for C_24_H_28_N_2_O (M^+^) 360.2202 (0.6 ppm). 

***(-)-(1S,2S,3R)-N,N-Diethyl-2-methyl-2-phenyl-3-(1H-pyrrolo[2,3-b]pyridin-1-yl)cyclopropane-1-carboxamide*** (**23acj**). Compound was obtained according to typical procedure B from 62.1 mg (0.201 mmol, 1.00 equiv.) of (1*R*,2*S*)-1-bromo-*N*,*N*-diethyl-2-methyl-2-phenylcyclopropane-1-carboxamide (**20ac**), employing 7-azaindole (**25j**) (71.0 mg, 0.600 mmol, 3.00 equiv.) as pronucleophile. Chromatographic purification afforded 35.4 mg (0.102 mmol, 50%) as a colorless solid, mp: 113.2–114.0 °C, R_ƒ_ 0.42 (hexanes/EtOAc 2:1), [α]^25^_D_ = −5.2° (c 0.669, CH_2_Cl_2_). dr 20:1. ^1^H NMR (500 MHz, CDCl_3_) *δ*_H_ 8.38 (dd, ^3^*J*_H,H_ = 4.7, ^4^*J*_H,H_ = 1.5 Hz, 1H), 7.90 (dd, ^3^*J*_H,H_ = 7.8 Hz, 1H), 7.65-7.51 (m, 2H), 7.34 (t, ^3^*J*_H,H_ = 7.7 Hz, 2H), 7.25 (d, ^3^*J*_H,H_ = 7.5 Hz, 1H), 7.22 (d, ^3^*J*_H,H_ = 3.6 Hz, 1H), 7.08 (dd, ^3^*J*_H,H_ = 7.8, 4.7 Hz, 1H), 6.47 (d, ^3^*J*_H,H_ = 3.5 Hz, 1H), 4.72 (d, ^3^*J*_H,H_ = 4.3 Hz, 1H), 3.85 (dq, ^2^*J*_H,H_ = 14.4, ^3^*J*_H,H_ = 7.1 Hz, 1H), 3.78-3.62 (m, 1H), 3.43 (dq, ^2^*J*_H,H_ = 14.3, ^3^*J*_H,H_ = 7.1 Hz, 1H), 2.97 (dq, ^2^*J*_H,H_ = 14.0, ^3^*J*_H,H_ = 7.0 Hz, 1H), 2.73 (d, ^3^*J*_H,H_ = 4.3 Hz, 1H), 1.40 (t, ^3^*J*_H,H_ = 7.1 Hz, 3H), 1.26 (s, 3H), 0.97 (t, ^3^*J*_H,H_ = 7.1Hz, 3H); ^13^C NMR (126 MHz, CDCl_3_) *δ*_C_ 166.9, 149.4, 143.6 (+), 141.1, 128.9 (+, 2C), 128.6 (+), 128.6 (+, 2C), 128.3 (+), 127.1 (+), 120.9 (+), 116.3 (+), 100.0 (+), 42.7 (+), 42.1 (+), 40.4 (-), 37.5, 33.3 (+), 22.6 (+), 14.9 (+), 12.8 (+); FT IR (KBr, cm^−1^): 2972, 2929, 1733, 1637, 1479, 1460, 1448, 1433, 1377, 1265, 1220, 1143, 1097, 1070, 952, 846, 796, 775, 763, 723, 702, 624, 598; HRMS (TOF ES): found 348.2076, calculated for C_22_H_26_N_3_O (M+H)^+^ 348.2070 (1.7 ppm).

***(+)-(1S,2S,3R)-N,N-Diethyl-2-methyl-3-(methyl(phenyl)amino)-2-phenylcyclopropane-1-carboxamide*** (**23ack**). Compound was obtained according to typical procedure B from 62.0 mg (.201 mmol, 1.00 equiv.) of (1*R*,2*S*)-1-bromo-*N*,*N*-diethyl-2-methyl-2-phenylcyclopropane-1-carboxamide (**20ac**), employing *N*-methylaniline (**25k**) (65.0 μL, 0.600 mmol, 3.00 equiv.) as pronucleophile. Chromatographic purification afforded 36.4 mg (0.110 mmol, 55.0%) as yellow oil, R*_f_* 0.22 (hexanes/EtOAc 5:1), dr 3:1 ^1^H NMR (500 MHz, CDCl_3_) *δ*_H_ 7.43–7.12 (m, 7H), 6.95 (d, ^3^*J*_H,H_ = 7.9 Hz, 2H), 6.79 (d, ^3^*J*_H,H_ = 7.3 Hz, 3H), 3.78 (d, ^3^*J*_H,H_ = 4.4 Hz, 1H), 3.68–3.55 (m, 2H), 3.19 (dq, ^2^*J*_H,H_ = 14.7, ^3^*J*_H,H_ = 7.2 Hz, 1H), 3.11 (s, 3H), 2.79 (dq, ^2^*J*_H,H_ = 14.7, ^3^*J*_H,H_ = 7.2 Hz, 1H), 1.97 (d, ^3^*J*_H,H_ = 4.4 Hz, 1H), 1.62 (s, 3H), 1.17 (t, ^3^*J*_H,H_ = 7.1 Hz, 3H), 0.79 (t, ^3^*J*_H,H_ = 7.1 Hz, 3H); ^13^C NMR (126 MHz, CDCl_3_) *δ*_C_ 167.2, 151.0, 141.0, 129.2 (+, 2C), 128.5 (+, 2C), 127.8 (+, 2C), 126.8 (+), 118.1, 114.8 (+, 2C), 48.5 (+), 41.6 (-), 40.4 (+), 39.8 (-), 38.3, 36.9 (+), 21.0 (+), 14.6 (+), 12.6 (+); FT IR (KBr, cm^−1^): 2972, 2929, 1639, 1598, 1500, 1479, 1444, 1433, 1379, 1305, 1220, 1143, 1116, 1029, 950, 904, 698, 611; HRMS (TOF ES): found 337.2281, calculated for C_22_H_29_N_2_O (M+H)^+^ 3370.2280 (0.3 ppm). [α]^25^_D_ = +33.1° (c 0.366, CH_2_Cl_2_).

***(+)-(1S,2S,3R)-N,N-Diethyl-3-(ethyl(phenyl)amino)-2-methyl-2-phenylcyclopropane-1-carboxamide*** (**23acl**). Compound was obtained according to typical procedure B from 62.0 mg (0.201 mmol, 1.00 equiv.) of (1*R*,2*S*)-1-bromo-*N*,*N*-diethyl-2-methyl-2-phenylcyclopropane-1-carboxamide (**20ac**), employing *N*-ethylaniline (**25l**) (75.0 μL, 0.600 mmol, 3.00 equiv.) as pronucleophile. Chromatographic purification afforded 44.8 mg (0.128 mmol, 64%) as a yellow oil, R_ƒ_ 0.31 (hexanes/EtOAc 5:1), [α]^25^_D_ = +11.6° (c 0.160, CH_2_Cl_2_). dr 13:1. ^1^H NMR (500 MHz, CDCl_3_) *δ*_H_ 7.51–7.14 (m, 7H), 7.01–6.89 (m, 2H), 6.76 (t, ^3^*J*_H,H_ = 7.3 Hz, 1H), 3.84 (d, ^3^*J*_H,H_ = 4.6 Hz, 1H), 3.70–3.42 (m, 4H), 3.14 (dq, ^2^*J*_H,H_ = 14.3, ^3^*J*_H,H_ = 7.1 Hz, 1H), 2.84 (dq, ^2^*J*_H,H_ = 13.9, ^3^*J*_H,H_ = 7.0 Hz, 1H), 1.95 (d, ^3^*J*_H,H_ = 4.6 Hz, 1H), 1.60 (s, 3H), 1.21 (t, ^3^*J*_H,H_ = 7.0 Hz, 3H), 1.14 (t, ^3^*J*_H,H_ = 7.1 Hz, 3H), 0.76 (t, ^3^*J*_H,H_ = 7.1 Hz, 3H); ^13^C NMR (126 MHz, CDCl_3_) *δ*_C_ 167.3, 149.2, 141.0, 129.2 (+, 2C), 128.4 (+, 2C), 127.6 (+, 2C), 126.7 (+, 2C), 118.0 (+), 115.7 (+), 46.3 (+), 46.2 (-), 41.5 (-), 39.7 (-), 36.9 (+), 14.5 (+), 12.5 (+), 11.2 (+); FT IR (KBr, cm^−1^): 3085, 2972, 2358, 1637, 1598, 1498, 1458, 1444, 1434, 1377, 1259, 1143, 1080, 831, 752, 696, 613; HRMS (TOF ES): found 350.2358, calculated for C_23_H_30_N_2_O (M^+^) 350.2358 (0.0 ppm). 

***(+)-(1S,2S,3R)-N,N-Diethyl-3-((4-fluorophenyl)(methyl)amino)-2-methyl-2-phenylcyclopropane-1-carboxamide*** (**23acm**). Compound was obtained according to typical procedure B from 59.8 mg (0.193 mmol, 1.00 equiv.) of (1*R*,2*S*)-1-bromo-*N*,*N*-diethyl-2-methyl-2-phenylcyclopropane-1-carboxamide (**20ac**), employing 4-fluoro-*N*-methylaniline (**25m**) (72.2 μL, 0.600 mmol, 3.11 equiv.) as pronucleophile. Chromatographic purification afforded 41.6 mg (0.118 mmol, 61%) of the title compound as a yellow oil, R*_f_* 0.25 (hexanes/EtOAc 4:1), [α]^25^_D_ = +35.4° (c 0.362, CH_2_Cl_2_). dr 3:1. ^1^H NMR (500 MHz, CDCl_3_) *δ*_H_ 7.35–7.27 (m, 4H), 7.21 (ddd, ^3^*J*_H,F_ = 5.0 Hz, ^3^*J*_H,H_ = 4.5, 1.9 Hz, 1H), 7.00–6.84 (m, 4H), 3.71 (d, ^3^*J*_H,H_ = 4.4 Hz, 1H), 3.67–3.55 (m, 2H), 3.18 (dd, ^2^*J*_H,H_ = 14.7, ^3^*J*_H,H_ = 7.2 Hz, 1H), 3.08 (s, 3H), 2.85 (dd, ^2^*J*_H,H_ = 13.6, ^3^*J*_H,H_ = 7.0 Hz, 1H), 1.91 (d, ^3^*J*_H,H_ = 4.4 Hz, 1H), 1.62 (s, 3H), 1.16 (t, ^3^*J*_H,H_ = 7.2 Hz, 3H), 0.79 (t, ^3^*J*_H,H_ = 7.1 Hz, 3H); ^13^C NMR (126 MHz, CDCl_3_) *δ*_C_ 167.1, 156.3 (d, ^1^*J*_C,F_ = 236.5 Hz), 147.5 (d, ^4^*J*_C,F_ = 1.9 Hz), 140.8, 128.4 (+, 2C), 127.6 (+, 2C), 126.7 (+), 116.0 (d, ^2^*J*_C,F_ = 7.4 Hz, +, 2C), 115.4 (d, ^3^*J*_C,F_ = 22.0 Hz, +, 2C), 48.8 (+), 41.5 (-), 41.1 (+), 39.7 (-), 38.1 (+), 36.8(+), 20.8 (+), 14.4 (+), 12.5 (+); FT IR (KBr, cm^−1^): 2974, 2873, 1635, 1510, 1479, 1458, 1446, 1379, 1263, 1224, 1143, 1022, 825, 763, 698; HRMS (TOF ES): found 353.2028, calculated for C_22_H_26_FN_2_O (M-H)^+^ 353.2029 (0.3 ppm). 

***(+)-(1R,2R,3S)-3-(Benzyloxy)-N-(tert-butyl)-2-ethyl-2-phenylcyclopropane-1-carboxamide*** (**23baf**). Compound was obtained according to typical procedure B from 72.2 mg (0.223 mmol, 1.00 equiv) of (1*S*,2*R*)-1-bromo-*N*-(*tert*-butyl)-2-ethyl-2-phenylcyclopropane-1-carboxamide (**20ba**), employing benzyl alcohol (**25f**) (64.9 μL, 0.624 mmol, 2.80 equiv.) as pronucleophile. Chromatographic purification afforded 49.0 mg (0.139 mmol, 70%) of the title compound as a colorless solid, mp: 136.2–137.1 °C, R_ƒ_ 0.23 (hexanes/EtOAc 9:1), [α]^25^_D_ = +50.0° (c 0.11, CH_2_Cl_2_). dr 39:1. Spectral properties of this material were identical to those reported for the racemic compound [67]. 

***(-)-(1R,2R,3S)-3-(Allyloxy)-N-(tert-butyl)-2-methyl-2-(p-tolyl)cyclopropane-1-carboxamide*** (**23cad**). Compound was obtained according to typical procedure B from 65.0 mg (0.200 mmol, 1.00 equiv.) of (1*S*,2*R*)-1-bromo-*N*-(*tert*-butyl)-2-methyl-2-(*p*-tolyl)cyclopropane-1-carboxamide (**20ca**), employing allyl alcohol (**25e**) (41.0 μL, 0.600 mmol, 3.00 equiv) as pronucleophile. Chromatographic purification afforded 42.0 mg (0.146 mmol, 73%) of the title compound as a colorless solid, mp: 120.6–122.7 °C, R*_f_* 0.38 (hexanes/EtOAc 5:1), [α]^25^_D_ = −21.8° (c 0.16, CH_2_Cl_2_). dr 50:1. ^1^H NMR (500 MHz, CDCl_3_) *δ*_H_ 7.12 (d, ^3^*J*_H,H_ = 8.1 Hz, 2H), 7.08 (d, ^3^*J*_H,H_ = 7.9 Hz, 2H), 5.99 (ddt, ^3^*J*_H,H_ = 16.2, 10.5, 5.7 Hz, 1H), 5.36 (dd, ^3^*J*_H,H_ = 17.2, ^2^*J*_H,H_ = 1.6 Hz, 1H), 5.24 (dd, ^3^*J*_H,H_ = 10.4, ^2^*J*_H,H_ = 1.4 Hz, 1H), 5.08 (br. s, 1H), 4.27–4.10 (m, 2H), 4.07 (d, ^3^*J* = 3.3 Hz, 1H), 2.29 (s, 3H), 1.65 (d, ^3^*J* = 3.3 Hz, 1H), 1.50 (s, 3H), 1.16 (s, 9H); ^13^C NMR (126 MHz, CDCl_3_) *δ*_C_ 168.1, 138.3, 136.4, 134.3 (+), 129.3 (+, 2C), 128.5 (+, 2C), 117.7 (-), 72.3 (-), 66.4 (+), 51.1, 37.0 (+), 37.0, 28.8 (+, 3C), 22.0 (+), 21.2 (+); FT IR (KBr, cm^−1^): 3301, 2966, 2923, 1643, 1546, 1515, 1454, 1226, 1145, 985, 925, 817; HRMS (TOF ES): found 300.1967, calculated for C_19_H_26_NO_2_ (M-H)^+^ 300.1964 (1.0 ppm). 

***(+)-(1R,2R,3S)-N-(tert-Butyl)-3-(2-methoxyethoxy)-2-methyl-2-(p-tolyl)cyclopropane-1-carboxamide*** (**23cae**). Compound was obtained according to typical procedure B from 62.5 mg (0.193 mmol, 1.00 equiv.) of (1*S*,2*R*)-1-bromo-*N*-(*tert*-butyl)-2-methyl-2-(*p*-tolyl)cyclopropane-1-carboxamide (**20ca**), employing 2-methoxyethanol (**25e**) (47.3 μL, 0.645 mmol, 3.22 equiv.) as pronucleophile. Chromatographic purification afforded 49.5 mg (0.155 mmol, 78%) of the title compound as a colorless solid, mp: 94.9–97.1 °C, R_ƒ_ 0.26 (hexanes/EtOAc 1:1), [α]^25^_D_ = +32.0° (c 0.05, CH_2_Cl_2_). dr > 100:1. Spectral properties of this material were identical to those reported for the racemic compound [67]. 

***(-)-(1R,2R,3S)-3-(Benzyloxy)-N-(tert-butyl)-2-methyl-2-(p-tolyl)cyclopropanecarboxamide*** (**23caf**) [67]. Compound was obtained according to typical procedure B from 130 mg (0.401 mmol, 1.00 equiv.) of (1*S*,2*R*)-1-bromo-*N*-(*tert*-butyl)-2-methyl-2-(*p*-tolyl)cyclopropane-1-carboxamide (**20ca**), employing benzyl alcohol (**25f**) (124 μL, 1.20 mmol, 3.00 equiv.) as pronucleophiles. The subsequent chromatographic purification afforded 129 mg (0.367 mmol, 92%) of the title compound as a colorless solid, mp: 139.8–140.6 ^o^C, R*_f_* 0.33 (hexanes/EtOAc 6:1), [α]^25^_D_ = –24.5° (c 1.10, CH_2_Cl_2_). dr 44:1. ^1^H NMR (500 MHz, CDCl_3_) *δ*_H_ 7.54–7.17 (m, 5H), 7.10 (q, ^3^*J*_H,H_ = 8.1 Hz, 4H), 5.03 (br. s, 1H), 4.87–4.53 (m, 2H), 4.12 (d, ^3^*J*_H,H_ = 3.3 Hz, 1H), 2.30 (s, 3H), 1.66 (d, ^3^*J*_H,H_ = 3.3 Hz, 1H), 1.54 (s, 3H), 1.16 (s, 9H); ^13^C NMR (126 MHz, CDCl_3_) *δ*_C_ 168.0, 138.3, 137.7, 136.4, 129.3 (+, 2C), 128.6 (+, 2C), 128.5 (+, 2C), 128.3 (+, 2C), 128.0 (+), 73.5 (-), 66.6 (+), 51.1, 37.1 (+), 28.8 (+, 3C), 22.1 (+), 21.2 (+); FT IR (KBr, cm^−1^): 3308, 3063, 3030, 2966, 2926, 2864, 1643, 1543, 1516, 1454, 1431, 1375, 1364, 1346, 1277, 1226, 1204, 1144, 1099, 987, 817, 750, 734, 698; HRMS (TOF ES): found 374.2098, calculated for C_23_H_29_NO_2_ (M+Na)^+^ 374.2096 (0.5 ppm). 

***(+)-(1R,2R,3S)-N-(tert-Butyl)-2-methyl-3-(1H-pyrrol-1-yl)-2-(p-tolyl)cyclopropane-1-carboxamide*** (**23cag**). Compound was obtained according to typical procedure B from 64.6 mg (0.199 mmol, 1.00 equiv.) of (1*S*,2*R*)-1-bromo-*N*-(*tert*-butyl)-2-methyl-2-(*p*-tolyl)cyclopropane-1-carboxamide (**20ca**), employing pyrrole (**25g**) (42.0 μL, 0.600 mmol, 3.00 equiv.) as pronucleophile. Chromatographic purification afforded 52.0 mg (0.168 mmol, 84%) of the title compound as a colorless solid, mp: 208.0–210.1 ^o^C, R*_f_* 0.31 (hexanes/EtOAc 6:1), [α]^25^_D_ = +46.5° (c 0.142, CH_2_Cl_2_). dr > 99:1. ^1^H NMR (500 MHz, CDCl_3_) *δ*_H_ 7.22 (d, ^3^*J*_H,H_ = 8.0 Hz, 2H), 7.14 (d, ^3^*J*_H,H_ = 7.9 Hz, 2H), 6.77 (t, ^3^*J*_H,H_ = 2.1 Hz, 2H), 6.19 (t, ^3^*J*_H,H_ = 2.1 Hz, 2H), 5.35 (br. s, 1H), 4.31 (d, ^3^*J*_H,H_ = 4.2 Hz, 1H), 2.32 (s, 3H), 2.17 (d, ^3^*J*_H,H_ = 4.2 Hz, 1H), 1.26 (s, 3H), 1.22 (s, 9H); ^13^C NMR (126 MHz, CDCl_3_) *δ*_C_ 166.9, 137.7, 137.0, 129.5 (+, 2C), 128.4 (+, 2C), 121.6 (+, 2C), 108.6 (+, 2C), 51.6, 45.0 (+), 37.1, 36.7 (+), 28.8 (+, 3C), 23.1 (+), 21.3 (+); FT IR (KBr, cm^−1^): 3319, 2966, 2925, 1645, 1546, 1539, 1492, 1454, 1361, 1265, 1224, 1093, 1066, 981, 817, 721, 700; HRMS (TOF ES): found 309.1974, calculated for C_20_H_25_N_2_O (M-H)^+^ 309.1967 (2.3 ppm). 

***(+)-(1R,2R,3S)-N-(tert-Butyl)-2-methyl-3-(1H-pyrazol-1-yl)-2-(p-tolyl)cyclopropane-1-carboxamide*** (**23cah**): Compound was obtained according to typical procedure B from 65.0 mg (0.200 mmol, 1.00 equiv.) of (1*S*,2*R*)-1-bromo-*N*-(*tert*-butyl)-2-methyl-2-(*p*-tolyl)cyclopropane-1-carboxamide (**20ca**), employing pyrazole (**25h**) (41.0 mg, 0.600 mmol, 3.00 equiv) as pronucleophiles. The subsequent chromatographic purification afforded 47.0 mg (0.158 mmol, 79%) of the title compound as a colorless solid, mp: 147.9–148.5 °C, R*_f_* 0.35 (CH_2_Cl_2_/MeOH 20:1), [α]^25^_D_ = +40.6° (c 0.35, CH_2_Cl_2_). dr 15:1. ^1^H NMR (500 MHz, CDCl_3_) *δ*_H_ 7.51 (d, ^3^*J*_H,H_ = 2.1 Hz, 1H), 7.47 (d, ^3^*J*_H,H_ = 1.2 Hz, 1H), 7.17 (d, ^3^*J*_H,H_ = 7.9 Hz, 2H), 7.05 (d, ^3^*J*_H,H_ = 7.9 Hz, 2H), 6.25 (t, ^3^*J*_H,H_ = 1.9 Hz, 1H), 5.69 (br. s, 1H), 4.46 (d, ^3^*J*_H,H_ = 4.0 Hz, 1H), 2.61–2.45 (m, 1H), 2.24 (s, 3H), 1.16 (s, 9H), 1.12 (s, 3H); ^13^C NMR (126 MHz, CDCl_3_) *δ*_C_ 166.5, 139.4 (+), 137.6, 136.8, 130.6 (+), 129.4 (+, 2C), 128.4 (+, 2C), 106.2 (+), 51.6, 47.1 (+), 37.4, 35.8 (+), 28.8 (+, 3C), 22.7 (+), 21.3 (+); FT IR (KBr, cm^−1^): 3306, 2964, 2925, 1649, 1544, 1452, 1392, 1274, 1224, 1089, 1047, 820, 752, 615; HRMS (TOF ES): found 312.2081, calculated for C_19_H_26_N_3_O (M+H)^+^ 312.2076 (1.6 ppm). 

***(-)-(1R,2R,3S)-3-(Allyloxy)-N-(tert-butyl)-2-methyl-2-(naphthalen-2-yl)cyclopropane-1-carboxamide*** (**23dad**). Compound was obtained according to typical procedure B from 69.9 mg (0.194 mmol, 1.00 equiv.) of (1*S*,2*R*)-1-bromo-*N*-(*tert*-butyl)-2-methyl-2-(naphthalen-2-yl)cyclopropane-1-carboxamide (**20da**), employing allyl alcohol (**25d**) (40.8 μL, 0.600 mmol, 3.09 equiv.) as pronucleophile. Chromatographic purification afforded 57.9 mg (0.172 mmol, 88%) of the title compound as a colorless solid, mp: 122.2–125.1 °C, R*_f_* 0.26 (hexanes/EtOAc 4:1), [α]^25^_D_ = −7.87° (c 0.178, CH_2_Cl_2_). dr > 99:1. ^1^H NMR (500 MHz, CDCl_3_) *δ*_H_ 7.85–7.72 (m, 3H), 7.69 (s, 1H), 7.50–7.36 (m, 2H), 7.35 (dd, ^3^*J*_H,H_ = 8.4, ^4^*J*_H,H_ = 1.7 Hz, 1H), 6.04 (ddt, ^3^*J*_H,H_ = 17.2, 10.5, 5.7 Hz, 1H), 5.41 (dd, ^3^*J*_H,H_ = 17.2, ^2^*J*_H,H_ = 1.6 Hz, 1H), 5.27 (dd, ^3^*J*_H,H_ = 10.4, ^2^*J*_H,H_ = 1.4 Hz, 1H), 5.17 (br. s, 1H), 4.34–4.05 (m, 2H), 4.22 (d, ^3^*J*_H,H_ = 3.3 Hz, 1H), 1.75 (d, ^3^*J*_H,H_ = 3.3 Hz, 1H), 1.59 (s, 3H), 1.13 (s, 9H); ^13^C NMR (126 MHz, CDCl_3_) *δ*_C_ 167.9, 139.0, 134.3 (+), 133.6, 132.6, 128.2 (+), 127.8 (+), 127.8 (+), 127.3 (+), 127.0 (+), 126.1 (+), 125.7 (+), 117.8 (-), 72.4 (-), 66.5 (+), 51.2, 37.5, 37.1 (+), 28.8 (+, 3C), 22.0 (+); FT IR (KBr, cm^−1^): 3319, 2966, 2925, 1643, 1542, 1454, 1361, 1269, 1226, 1147, 1128, 1087, 1062, 1041, 985, 923, 856, 817, 744, 667; HRMS (TOF ES): found 338.2119, calculated for C_22_H_28_NO_2_ (M+H)^+^ 338.2120 (0.3 ppm). 

***(-)-(1R,2R,3S)-N-(tert-Butyl)-3-(2-methoxyethoxy)-2-methyl-2-(naphthalen-2-yl)cyclopropane-1-carboxamide*** (**23dae**). Compound was obtained according to typical procedure B from 71.8 mg (0.199 mmol, 1.00 equiv.) of (1*S*,2*R*)-1-bromo-*N*-(*tert*-butyl)-2-methyl-2-(naphthalen-2-yl)cyclopropane-1-carboxamide (**20da**), employing 2-methoxyethanol (**25e**) (47.3 μL, 0.600 mmol, 3.02 equiv.) as pronucleophile. Chromatographic purification afforded 53.8 mg (0.151 mmol, 76%) of the title compound as a colorless solid, mp: 143.2–144.2 °C, R*_f_* 0.25 (hexanes/EtOAc 2:1), [α]^25^_D_ = −6.1° (c 0.214, CH_2_Cl_2_). dr > 99:1. ^1^H NMR (500 MHz, CDCl_3_) *δ*_H_ 7.86–7.65 (m, 4H), 7.54–7.32 (m, 3H), 5.19 (br. s, 1H), 4.23 (d, ^3^*J*_H,H_ = 3.3 Hz, 1H), 3.96–3.77 (m, 2H), 3.73–3.60 (m, 2H), 3.44 (s, 3H), 2.17 (s, 3H), 1.76 (d, ^3^*J*_H,H_ = 3.3 Hz, 1H), 1.12 (s, 9H); ^13^C NMR (126 MHz, CDCl_3_) *δ*_C_ 167.9, 139.0, 133.6, 132.6, 128.2 (+), 127.8 (+), 127.3 (+), 127.0 (+), 126.0 (+), 125.7 (+), 71.9 (-), 70.5 (-), 67.0 (+), 59.3 (+), 51.2, 37.6, 37.0 (+), 31.1 (+), 28.8 (+, 3C), 21.9 (+); FT IR (KBr, cm^−1^): 3323, 2966, 2871, 1645, 1541, 1454, 1390, 1363, 1269, 1226, 1151, 1124, 956, 856, 817, 742, 667; HRMS (TOF ES): found 356.2225, calculated for C_22_H_30_NO_3_ (M+H) 356.2226 (0.3 ppm). 

***(+)-(1R,2R,3S)-N-(tert-Butyl)-2-methyl-2-(naphthalen-2-yl)-3-(1H-pyrrol-1-yl)cyclopropane-1-carboxamide*** (**2****3dag**). Compound was obtained according to typical procedure B from 71.4 mg (0.198 mmol, 1.00 equiv.) of (1*S*,2*R*)-1-bromo-*N*-(*tert*-butyl)-2-methyl-2-(naphthalen-2-yl)cyclopropane-1-carboxamide (**20da**), employing pyrrole (**25g**) (42.0 μL, 0.606 mmol, 3.06 equiv.) as pronucleophile. Chromatographic purification afforded 51.9 mg (0.149 mmol, 75%) of the title compound as a colorless solid, mp: 181.4–183.2 ^o^C, R*_f_* 0.23 (hexanes/EtOAc 6:1), [α]^25^_D_ = +15.6° (c 0.096 CH_2_Cl_2_). dr 81:1. ^1^H NMR (500 MHz, CDCl_3_) *δ*_H_ 8.01–7.67 (m, 4H), 7.57–7.35 (m, 3H), 6.85 (t, ^3^*J*_H,H_ = 2.1 Hz, 2H), 6.23 (t, ^3^*J*_H,H_ = 2.1 Hz, 2H), 5.44 (br., s, 1H), 4.46 (d, ^3^*J*_H,H_ = 4.1 Hz, 1H), 2.28 (d, ^3^*J*_H,H_ = 4.2 Hz, 1H), 1.36 (s, H), 1.19 (s, 9H); ^13^C NMR (126 MHz, CDCl_3_) *δ*_C_ 166.7, 138.3, 133.6, 132.7, 128.5 (+), 127.8 (+), 127.8 (+), 127.3 (+), 126.7 (+), 126.3 (+), 126.0 (+), 121.7 (+, 2C), 108.7 (+, 2C), 51.7, 45.2 (+), 37.7, 36.8 (+), 28.8 (+, 3C), 23.0 (+); FT IR (KBr, cm^−1^): 3305, 2968, 1650, 1548, 1492, 1454, 1392, 1265, 1224, 1132, 1091, 1064, 981, 854, 817, 721, 680, 657; HRMS (TOF ES): found 346.2047, calculated for C_23_H_26_N_2_O (M^+^) 346.2045 (0.6 ppm). 

***(1R*,2R*,3S*)-N-(tert-butyl)-2-ethyl-2-phenyl-3-(1H-pyrrol-1-yl)cyclopropane-1-carboxamide*** (**23aag**). This compound was obtained according to procedure B, employing pyrrole (42 μL, 0.60 mmol, 3.0 equiv.) as pronucleophile. The reaction mixture was stirred overnight at 40 °C, and GC analysis showed incomplete conversion (75% based on starting material) and dr 17:1; after heating at 80 °C for 30 min, the reaction was complete. The subsequent chromatographic purification afforded 35 mg (0.114 mmol, 57%) of the title compound as a white solid, mp: 177.5–120.0 ^o^C, R*_f_* 0.29 (hexanes/EtOAc 6:1), dr 14:1. ^1^H NMR (500 MHz, CDCl_3_) *δ*_H_ 7.44–7.07 (m, 5H), 6.72 (t, ^3^*J*_H,H_ = 2.1 Hz, 2H), 6.12 (t, ^3^*J*_H,H_ = 2.1 Hz, 2H), 5.28 (br. s, 1H), 4.25 (d, ^3^*J*_H,H_ = 4.3 Hz, 1H), 2.12 (d, ^3^*J*_H,H_ = 4.3 Hz, 1H), 1.45–1.32 (m, 2H), 1.16 (s, 9H), 0.71 (t, *J*_H,H_ = 7.4 Hz, 3H); ^13^C NMR (126 MHz, CDCl_3_) *δ*_C_ 166.9, 138.6, 129.7 (+, 2C), 128.4 (+, 2C), 127.4 (+), 121.7 (+, 2C), 108.6 (+, 2C), 51.6, 46.3 (+), 43.0, 35.1 (+), 28.8 (+, 3C), 28.5 (-), 11.3 (+); FT IR (KBr, cm^−1^): 3317, 3060, 2968, 2931, 1647, 1545, 1492, 1446, 1263, 1224, 721, 698; HRMS (TOF ES): found 309.1968, calculated for C_20_H_25_N_2_ONa (M-H) 342.2045 (0.6 ppm).

***(+)-(1R,2R,3S)-N-(tert-Butyl)-3-(ethyl(phenyl)amino)-2-methyl-2-(naphthalen-2-yl)cyclopropane-1-carboxamide*** (**23dal**). Compound was obtained according to typical procedure B from 39.1 mg (0.109 mmol, 1.00 equiv.) of (1*S*,2*R*)-1-bromo-*N*-(*tert*-butyl)-2-methyl-2-(naphthalen-2-yl)cyclopropane-1-carboxamide (**20da**), employing *N*-ethylaniline (**25l**) (40 μL, 0.318 mmol, 2.92 equiv.) as pronucleophile. Chromatographic purification afforded 29.8 mg (0.074 mmol, 68%) of the title compound as a colorless solid, m.p. 132.3–135.1 °C, R*_f_* 0.31 (hexanes/EtOAc 6:1), [α]^25^_D_ = +10.8° (c 0.074, CH_2_Cl_2_). dr > 99:1. ^1^H NMR (500 MHz, CDCl_3_) *δ*_H_ 7.97–7.66 (m, 4H), 7.55–7.39 (m, 3H), 7.27–7.19 (m, 2H), 7.06–6.93 (m, 2H), 6.80 (t, ^3^*J*_H,H_ = 7.3 Hz, 1H), 5.18 (br. s, 1H), 3.71 (d, ^3^*J*_H,H_ = 4.4 Hz, 1H), 3.77–3.64 (m, 1H), 3.61–3.48 (m, 1H), 1.73 (d, ^3^*J*_H,H_ = 4.4 Hz, 1H), 1.61 (s, 3H), 1.25 (t, ^3^*J*_H,H_ = 7.0 Hz, 3H), 1.10 (s, 9H); ^13^C NMR (126 MHz, CDCl_3_) *δ*_C_ 167.8, 149.2, 139.2, 133.6, 132.5, 129.2 (+, 2C), 128.3 (+), 127.9 (+), 127.8 (+), 126.9 (+), 126.8 (+), 126.2 (+), 125.8 (+), 118.3 (+), 116.0 (+), 51.3, 46.8 (+), 46.4 (-), 39.2 (+), 39.0, 28.8 (+, 3C), 21.9 (+), 11.2 (+); FT IR (KBr, cm^−1^): 2968, 2358, 1645, 1595, 1531, 1498, 1454, 1366, 1255, 1188, 817, 742, 692; HRMS (TOF ES): found 399.2438, calculated for C_27_H_31_N_2_O (M-H)^+^ 399.2436 (0.5 ppm).

## 4. Conclusions

In conclusion, a highly efficient method for the assembly of tetrasubstituted chiral non-racemic cyclopropanes with all three asymmetric carbons in the strained ring was demonstrated. This method utilizes a “dual-control” strategy, which was successfully employed for the highly diastereoselective addition of the nucleophilic species to in situ generated enantiomerically enriched cyclopropenes. The chiral integrity of the starting material was translated to the product via the sequential installation of two stereogenic centers that were efficiently controlled by steric and thermodynamic effects. Alkoxides, as well as nitrogen-based nucleophiles (azoles and anilines), were used to access the homochiral derivatives of cyclopropyl ethers and cyclopropylamines. These reactions proceeded smoothly, affording unusually conformationally constrained amide derivatives of densely substituted enantiomerically enriched β-amino acids possessing three contiguous stereogenic carbon atoms. It should be also pointed out that one of these centers is an all-carbon-substituted quaternary stereocenter, the installation of which, by traditional methods, represents a long-standing challenge.

## Data Availability

Appendix A data include NMR spectral charts.

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
