# Peer review of "Preparation of Chiral Enantioenriched Densely Substituted Cyclopropyl Azoles, Amines, and Ethers via Formal SN2′ Substitution of Bromocylopropanes"

_molecules, 2022, doi:10.3390/molecules27207069_

Round 1

Reviewer 1 Report

The paper is written very well, it is clear and concise. The authors explain very well in this paper about synthesis of enantiopure cyclopropyl ethers, amines.

I have few comments included for the betterment of the paper.

1.      Please include this reference in diastereoselective synthesis of cyclopropanation through 1,3-ring closure:

-Directed, Remote Dirhodium C(sp3)-H Functionalization, Desaturative Annulation, and Desaturation

(https://doi.org/10.1021/jacs.2c07427)

2.     Can authors comment on why secondary and tertiary alcohols are not tolerate in these reactions.

3.     Epoxides and oxetanes are pseudo-oxygen nucleophiles. Instead of alcohol as nucleophile if use pseudo-oxygen nucleophiles, the reaction can be tolerated?

4.     Page-3, in last para bromocyclopropanes number should be 20.

5.     Please authors comment on 23ddl product. Why is this substrate not tolerate?

Author Response

Reviewer 1

Comments and Suggestions for Authors

The paper is written very well, it is clear and concise. The authors explain very well in this paper about synthesis of enantiopure cyclopropyl ethers, amines. 

I have few comments included for the betterment of the paper. 

  1. Please include this reference in diastereoselective synthesis of cyclopropanation through 1,3-ring closure:

-Directed, Remote Dirhodium C(sp3)-H Functionalization, Desaturative Annulation, and Desaturation

(https://doi.org/10.1021/jacs.2c07427)

Response: The citation of this paper was added. 

  1. Can authors comment on why secondary and tertiary alcohols are not tolerate in these reactions.

Response: There is an issue of competing nucleophilicity of tert-butoxide employed as a base.  In order to obtain good chemo-selectivity, we have to employ highly nucleophilic primary alcohols.  In reactions involving secondary and tertiary alcohols notable amounts of tert-butoxide adduct impurities are formed, which are very hard to separate from the target products.  Such reactions were tested, but they did not provide products with reasonable yields or analytical data suitable for publishing.      

  1. Epoxides and oxetanes are pseudo-oxygen nucleophiles. Instead of alcohol as nucleophile if use pseudo-oxygen nucleophiles, the reaction can be tolerated?

Response: We demonstrated previously (see Refs. 63, 64), that for successful cascade transformation involving both dehydrobromination and nucleophilic addition steps (i.e. formal nucleophilic substitution), it is very important to balance nucleophilicity and acid-base properties of the employed reactants.  It seems that pKa of the reactive pronucleophilic species should fall in a relatively narrow range.  This is probably why epoxides and oxetanes cannot be used.  At least we never were able to engage them in any of the related transformations.

  1. Page-3, in last para bromocyclopropanes number should be 20

Response: this typo was corrected.

  1. Please authors comment on 23ddl product. Why is this substrate not tolerate?

Response: We attribute this to decreased nucleophilicity of sterically demanding aniline bearing more bulky isopropyl substituent at nitrogen atom.  Short statement reflecting this idea was added to the text.    

Reviewer 2 Report

In this paper, Rubin et al describe the diastereoselective addition of O- and N-nucleophiles onto in situ generated cyclopropenes. High diastereoselectivity is observed in most cases, enabling the enantioselective preparation of chiral cyclopropyl ether and amines possessing three stereocenters.

Unfortunately, the obtaining of starting enantioenriched bromocyclopropane derivaitves relies on the use of a chiral auxiliary and a chiral resolution what limits the practical interest of the methodology.

This study is however interesting and deserves publication in Molecules.

Note: In the title, “SN2’” should be “SN2’”

Author Response

Reviewer 2

Comments and Suggestions for Authors

In this paper, Rubin et al describe the diastereoselective addition of O- and N-nucleophiles onto in situ generated cyclopropenes. High diastereoselectivity is observed in most cases, enabling the enantioselective preparation of chiral cyclopropyl ether and amines possessing three stereocenters.

Unfortunately, the obtaining of starting enantioenriched bromocyclopropane derivaitves relies on the use of a chiral auxiliary and a chiral resolution what limits the practical interest of the methodology.

This study is however interesting and deserves publication in Molecules.

Response: In this paper we did not claim the novel method for preparation of the starting materials, i.e. enantioenriched bromocyclopropane derivatives.  It should be pointed out, that such substrates are also available via asymmetric cyclopropanation using Co(III) complexes with chiral porphyrin ligands.  Such ligands, however are not commercially available, their synthesis is very tedious and expensive, which makes this method cost-prohibitive as compared to our large scale chiral resolution of diastereomeric salts with inexpensive natural alkaloids.  However, all these issues were already discussed in our publication in 2014, while now we focus on utization of this pool of chiral cyclopropane synthons.

Note: In the title, “SN2’” should be “SN2’”

Response: This issue was fixed.

Reviewer 3 Report

The comments regarding the paper submitted to Molecules with Manuscript ID molecules-1960386. The authors reported the preparation of enantiomerically enriched cyclopropyl derivatives with three asymmetric carbon in the strained ring by using “dual-control” strategy with moderate to good diastereoselectivity (up to 99%) and moderate to good yields (up to 92%). The manuscript bears good chemistry and is well-written. The chiral products were fully characterized. It is a nice work on chiral cyclopropyl for a challenge organic reaction and is worth to be published in Molecules. It seems that it contains some typical errors that should be corrected.

      In line 104: “… bromocyclopropanes 19 were converted …”  should be replaced with “… bromocyclopropanes 20 were converted …” 

      In line 118: “… chiral amides 23 for the generation…”  should be replaced with “… chiral amides 20 for the generation…” 

      Please add R and S configuration of 23aag in scheme 5

      In line 332: “dr>100:1 was correct?” 

Author Response

Reviewer 3

Comments and Suggestions for Authors

 The comments regarding the paper submitted to Molecules with Manuscript ID molecules-1960386. The authors reported the preparation of enantiomerically enriched cyclopropyl derivatives with three asymmetric carbon in the strained ring by using “dual-control” strategy with moderate to good diastereoselectivity (up to 99%) and moderate to good yields (up to 92%). The manuscript bears good chemistry and is well-written. The chiral products were fully characterized. It is a nice work on chiral cyclopropyl for a challenge organic reaction and is worth to be published in Molecules. It seems that it contains some typical errors that should be corrected.

  • In line 104: “… bromocyclopropanes 19 were converted …”  should be replaced with “… bromocyclopropanes 20 were converted …”  
  • In line 118: “… chiral amides 23for the generation…”  should be replaced with “… chiral amides 20 for the generation…”  
  • Please add R and S configuration of 23aagin scheme 5

Response: All these issues were fixed. 

  • In line 332: “dr>100:1 was correct?” 

Response: These notations simply mean that we were not able to detect any amounts of the second diastereomers in the corresponding experiments.  We added the relevant statement to the Chapter 3.1.